# Crosstalk between Circulating Tumor Cells and Plasma Proteins—Impact on Coagulation and Anticoagulation

**DOI:** 10.3390/cancers15113025

**Published:** 2023-06-01

**Authors:** Yuanyuan Wang, Stefan W. Schneider, Christian Gorzelanny

**Affiliations:** Department of Dermatology and Venereology, University Medical Center Hamburg-Eppendorf, Martinistrasse 52, 20246 Hamburg, Germany

**Keywords:** circulating cancer cell, cancer-associated thrombosis, plasma protein, coagulation, heparan sulfate

## Abstract

**Simple Summary:**

The interactions between circulating tumor cells (CTCs) and plasma proteins are critical for hematogenous metastasis, and understanding the molecular mechanisms behind these interactions can improve liquid-biopsy-based diagnostics and cancer therapies. This review summarizes recent literature on the surface molecules of CTCs that interact with coagulation proteins and their biological and clinical relevance. It also discusses future research directions to expand our knowledge of the CTC interactome, which can lead to the discovery of new molecular markers for diagnostics and additional targets for cancer therapies.

**Abstract:**

Cancer metastasis is a complex process. After their intravasation into the circulation, the cancer cells are exposed to a harsh environment of physical and biochemical hazards. Whether circulating tumor cells (CTCs) survive and escape from blood flow defines their ability to metastasize. CTCs sense their environment with surface-exposed receptors. The recognition of corresponding ligands, e.g., fibrinogen, by integrins can induce intracellular signaling processes driving CTCs’ survival. Other receptors, such as tissue factor (TF), enable CTCs to induce coagulation. Cancer-associated thrombosis (CAT) is adversely connected to patients’ outcome. However, cancer cells have also the ability to inhibit coagulation, e.g., through expressing thrombomodulin (TM) or heparan sulfate (HS), an activator of antithrombin (AT). To that extent, individual CTCs can interact with plasma proteins, and whether these interactions are connected to metastasis or clinical symptoms such as CAT is largely unknown. In the present review, we discuss the biological and clinical relevance of cancer-cell-expressed surface molecules and their interaction with plasma proteins. We aim to encourage future research to expand our knowledge of the CTC interactome, as this may not only yield new molecular markers improving liquid-biopsy-based diagnostics but also additional targets for better cancer therapies.

## 1. Introduction

Cancer cells which shed from a primary tumor and circulate in the bloodstream are referred to as circulating tumor cells (CTCs). The presence of CTCs in blood has been largely associated with cancer progression [1,2,3], and it serves as an indicator of the systemic dissemination of cancer, identifying patients at a higher risk of developing metastasis [4,5]. However, the effective number of CTCs is low, as less than 0.01% of the tumor cells exposed to blood can survive [6] and their half-life is estimated to be only around 20 min [7]. CTCs face numerous challenges in the blood, including immune surveillance, the coagulation system and mechanical stress [8]. Only CTCs that can survive the hostile blood environment can form metastases. Critical to this process is also the CTC’s ability to escape from the circulation system through attaching to the blood vessel wall and transmigrating into the perivascular tissues [9]. In addition to their survival and escape abilities, CTCs may also possess the capacity to promote tumor growth through inducing angiogenesis and other tumor supportive pathways once they extravasated into the surrounding tissue. Overall, the complex interplay between CTCs and the blood microenvironment is a crucial determinant of the metastatic potential of cancer cells.

Accumulating studies suggest that the success of the metastatic process is conditioned via the molecular crosstalk between tumor cells and the hostile blood microenvironment [10]. During the metastatic process, CTCs have the opportunity to interact tightly with myeloid-derived suppressor cells (MDSCs) [11], neutrophils [12], NK cells [13], macrophages, T cells [14,15] and platelets [16]. Depending on the cancer cell repertoire, these cell–cell interactions may result in CTC elimination but could also empower them to escape immune recognition. MDSCs interact with CTCs via jagged canonical Notch ligand 1 (JAG1) and neurogenic locus notch homolog protein 1 (Notch1) supporting the adhesion of the cancer cell to the endothelium and extravasation [17]. Neutrophils were shown to bind to vascular adhesion molecule 1 (VCAM1) on CTCs, which promoted CTC proliferation and metastasis [18]. The extravasation process is fueled by inflammatory cytokines released from neutrophils, which induce the expression of adhesion molecules at the endothelium [19]. Interestingly, the binding of neutrophils to CTCs is believed to prevent the attack of NK cells [12]. In case the function of NK cells is not quenched, they were able to clear CTCs. The physical contact between NK and CTC, which is required for the elimination of the tumor cell by granzyme B, death receptor signaling or perforin, is facilitated via the NK cell exposed lectin-like receptor K1 (KLRK1) sensing the major histocompatibility complex class 1 (MHC1) on CTCs [13]. Similar to NK cells, cytotoxic T cells are also able to eliminate CTCs. Immunological synapse formation between the MHC-exposed antigens on CTCs and T cell receptors initiates the killing of the cancer cells via death receptor ligand binding [20]. Additionally, CTCs have been reported to bind platelets [16], which are considered to be tumor-supportive. It has been suggested that CTCs form a shield of platelets around themselves to protect against the attack of NK cells [21,22]. The detailed crosstalk between CTCs and blood cells is beyond the scope of the present review. We refer to excellent other reviews [6,23]. In the present article we aim to focus on CTC receptors that enable interaction with plasma proteins.

Besides blood cells, the plasma contains thousands of different proteins [24,25]. Among all plasma proteins, 93% are albumin and globulins. Most of the remaining 7% consists of fibrinogen and other coagulation factors. Although the interaction between CTCs and plasma proteins is obvious, and potentially connected to the fate of the CTC, only a few studies addressing this topic are currently available. For example, the interaction between CTCs and the components of the coagulation system is of particular interest, as it may contribute to cancer-associated thrombosis (CAT) [26,27]. CAT is reported to be the second most common cause of death in patients with cancer [28,29] and was first identified by Trousseau in 1865. As a systemic hypercoagulable state found in many cancer patients, CAT can present as venous thromboembolism (VTE) [30], pulmonary embolism (PE) [31], arterial thromboembolism (ATE) [32] or microthrombosis. VTE is the most common form of CAT and refers to the formation of blood clots in the veins, usually in the legs, which can break loose and travel to the lungs as a PE. ATE refers to the formation of blood clots in the arteries, which can cause a heart attack or stroke. Microthrombosis is the formation of small clots in the microvasculature, leading to tissue ischemia and organ dysfunction.

Many studies have focused on the impact of the tumor microenvironment on systemic effects such as thrombocytosis [33] or neutrophil build immunothrombosis through releasing neutrophil extracellular traps (NETs) [34,35]. However, research investigating CAT from the perspective of CTCs is scarce, and although it is known that CTCs can express procoagulant proteins [30,36] and that they appear as active part of thrombi [26,27], it remains largely unknown how CTCs respond to the binding of plasma proteins and whether the corresponding cellular response promotes metastasis or further coagulation.

The interactions between CTCs and plasma proteins are potentially mediated by various cell surface receptors, such as tissue factor (TF), integrins, thrombomodulin (TM) and heparan sulfate (HS). These receptors can recognize and bind to different plasma proteins, influencing the fate of the CTCs. TF, for example, is a key factor in coagulation and recognizes FVIIa, while integrins can bind to fibrinogen and fibronectin in the context of platelet activation. TM, on the other hand, can counteract TF-promoted coagulation through directly interacting with thrombin. HS has the potential to promote or suppress coagulation via binding to various plasma proteins such as antithrombin (AT), fibrinogen or von Willebrand factor (vWF).

The interaction between CTCs and different plasma proteins is evidently important, but our current understanding of its biological and clinical significance is limited. In this review, we aim to highlight the relevance of this interaction and encourage further research on this topic. Delineation of these interactions is crucial to better understand tumor progression and to identify potential therapeutic targets. In the following sections, these interactions are described in detail.

## 2. Tissue Factor

TF, also known as coagulation Factor III, CD142 or thromboplastin, is a 47 kDa cell-surface, transmembrane protein of the class II cytokine receptor family. Its expression correlates with the frequency of CAT in cancer patients [28,29]. Under normal homeostatic conditions, TF is only exposed to the blood upon vascular injury, implicating endothelial damage and, thus, contact of the blood to the subendothelial TF-expressing smooth muscle cells. In response to this, TF interacts with calcium and other coagulation factors in the blood to trigger hemostasis.

In contrast to this hemostatic situation, various studies document that CTCs express TF and that they shed TF-positive microvesicles into circulation [37]. Interestingly, TF expression is not a specialty limited to a few cancer cells but has shown to be significantly expressed on the cellular surface of tumor cells of different tumor entities [38,39,40,41]. Functional TF expressed by CTCs can initiate the extrinsic coagulation cascade [42,43] through binding to FVII, which is then rapidly converted to FVIIa. The active TF-FVIIa complex catalyzes the activation of FX, which then forms the prothrombinase complex that converts prothrombin to thrombin. Thrombin cleaves fibrinogen into fibrin and activates platelets via protease-activated receptors (PARs), promoting the formation of hemostatic clots. Additionally, thrombin activates FVIII, FXIII and FV, amplifying the coagulation response. Overall, the expression of functional TF by CTCs highlights its significance in the pathophysiology of cancer and its potential role in promoting thrombosis. 

However, TF plays a vital role in tumor progression not only through initiating thrombosis. The TF-FVIIa complex at the CTC surface contributes to tumor cell proliferation, angiogenesis and metastasis [38,44] through activating PARs and mitogen-activated protein kinase (MAPK) pathway promoting, e.g., epithelial–mesenchymal transition (EMT) (Figure 1). EMT is a key process that enhances the ability of CTCs to disseminate as it drives vascular extravasation. Therefore, molecular markers of EMT and related signaling pathways may predict the ability of CTCs to metastasize [41,45,46,47,48]. In this context, it has been shown that the formation of the TF–FVIIa complex at the surface of breast cancer and lung carcinoma cells mediates PAR-2 signaling [49,50,51]. This results in MAPK pathway activation and further downstream effects such as the upregulation of cytokines and proangiogenic factors [52]. For the activation of the p44/42 MAPK pathway, as well as JAK2, p70/p85S6K and p90RSK, TF cytoplasmic domain is not necessary [53,54,55], but the extracellular domain which can bind FVIIa is required. Additionally, the TF–FVIIa-FXa complex-mediated activation of PAR-2 induced IL-8 expression and triggered cell migration [56]. Next to PAR-2, TF-mediated formation of thrombin was shown to activate PAR-1 in different tumor cells promoting, e.g., metastasis [57,58]. Interestingly, the prometastatic effect of PAR-1 requires co-expression with TF at the surface of the tumor cells, while the exclusive overexpression of PAR-1 was not sufficient [59].

## 3. Integrins

The binding of FVII to tumor cell surfaces via TF induces thrombin formation, which cleaves fibrinogen to fibrin and leads to blood clotting [60]. Additionally, another mechanism linking TF to metastasis is through the fibrinogen-dependent and platelet-dependent restriction in natural killer cell-mediated clearance of micrometastases [61]. Fibrinogen, which is one of the most abundant coagulation proteins, can directly bind to the surface of CTCs via integrins [62,63]. Integrins are transmembrane receptors ubiquitously expressed on cell surfaces and they play a critical role in regulating interactions between cells and their environment. These receptors are responsible for mediating cell–cell, cell–extracellular matrix and cell–protein interactions via responding to both intracellular and extracellular signals [64,65,66,67,68]. Integrins are obligate heterodimeric receptors consisting of an α and a β subunit. There are 18 α and 8 β subunits and various combinations comprise a family of 24 heterodimeric integrins members [62,69].

Most integrins can interact with multiple ligands [62]. Vice versa, one integrin ligand can interact with different integrins. Based on their structural similarity and ligand recognition patterns, integrins can be grouped into four classes: RGD-binding, collagen-binding, laminin-binding and leukocyte-specific integrin receptors [68]. The abbreviation RGD stands for the tripeptide sequence: Arg-Gly-Asp. Since the RGD peptide is the common feature of various plasmatic proteins [62,70,71], comprising, e.g., fibrinogen [72], fibronectin [73] and vWF [74,74,75,76], this review focuses exclusively on RGD-binding integrins. Most of the RGD-binding integrins are expressed at elevated levels in various cancer types, suggesting their crucial contribution to pathophysiological functions in cancer [77].

Plasma fibrinogen and its related fragments are ascribed to play an important role in the pathophysiology of tumor angiogenesis, invasion and metastasis [60,61]. It has been shown that a high pre-therapeutic plasma fibrinogen level independently predicts adverse events and the clinical outcome of patients with various cancer types [78,79,80,81]. Clinical studies have confirmed that elevated plasma fibrinogen levels are associated with poor outcome in patients with non-metastatic and metastatic renal cell carcinoma [82,83]. Though the β3 integrin family includes αIIbβ3 and αvβ3, often referred to as the fibrinogen receptor, fibrinogen can also bind to other integrins including α5β1 [84]. In lung cancer, the prevention of fibrinogen–integrin interaction via fibrinogen knockout promoted tumor growth and metastasis through serine/threonine kinase (also known as PKB) AKT signaling [85]. Fibrinogen binding to integrins α5β1 and αvβ3 promoted the activation of NF-κB and increased the expression of NF-κB-mediated inflammatory chemokines [86].

Fibronectin is a large, disulfide-linked dimeric glycoprotein that shares the same integrin binding receptor for αvβ3 and α5β1 with fibrinogen [62]. Fibronectin is present in a soluble form in plasma or as insoluble filaments deposited in the extracellular matrix [87]. During normal wound healing, the plasma fibronectin or insoluble fibrinogen filaments are released by damaged blood vessels and repair tissue through forming scaffolds with platelets [88]. Malignant tumors often develop at sites of chronic injury and permanent wounds, where the leaky vessels form fibrin clots around the tumoral lesion [89]. Hence, the formation of thrombotic clots via recruiting fibronectin to tumor cell surface integrins was thought to support CTC extravasation [90,91].

Under dynamic flow conditions, melanoma surface integrin αvβ3 has the unique ability to interact with immobilized fibrinogen and fibronectin and support cell arrest [92]. Previous studies also showed that integrin αIIbβ3 can catch blood-flowing fibrinogen in the context of platelet aggregation. This contributes to initial tumor cell arrest at the vascular endothelium during the early stage of tumor cell dissemination [93]. Lung cancer cell α5β1 integrin interacts with fibronectin, promoting cancer cell proliferation and metastasis [94]. However, direct interaction between fibrinogen and α5β1 integrins was proven only under static conditions. Under flow conditions, α5β1 integrins did not contribute to cancer cell arrest on immobilized fibrinogen [92]. The interaction of integrins with plasmatic proteins and its impact on tumor progression is schematically summarized in Figure 2.

## 4. Thrombomodulin

Although the formation of thrombin can promote the generation of fibrin and, thus, blood clotting, thrombin can also directly bind to tumor cell surface-exposed TM. Once bound to the surface, the protein C (PC) anti-coagulant pathway is switched on, representing an important negative feedback regulator of the coagulation cascade.

Mature human TM is a 557-amino-acid residue, type 1 transmembrane glycoprotein. It was discovered in the 1980s [95]. TM interacts with thrombin, building the TM-thrombin complex, which catalyzes the activation of the zymogen PC. The activated protein C (APC) together with protein S (PS) form the APC–PS complex, preventing further activation of FV and FXIII [96]. The PC anti-coagulant pathway switches thrombin from a pro-coagulant to an anti-coagulant enzyme.

PS is not only a cofactor of activated PC, but also a binding partner of Tissue Factor Pathway Inhibitor (TFPI) [97]. TFPI is the physiological inhibitor of the TF pathway of blood coagulation. The binding of PS to TFPI inhibits FXa. The TFPI/FXa complex binds to the complex of TF and FVIIa to form an inactive quaternary TFPI/FXa/TF/FVIIa complex. The protein-S-promoted activity of PC is further enhanced in the presence of calcium and phospholipids [98]. The anticoagulative activity of APC is inhibited by the Protein C Inhibitor (PCI), also known as Plasminogen Activator Inhibitor 3 (PAI3).

In the context of tumor progression, TM is supposed to have tumor-suppressive properties. Numerous preclinical and clinical studies indicate that the dampening of cell proliferation, invasion and metastasis is mediated by TM (Figure 3) [99,100,101]. A mouse model expressing a mutant form of TM with reduced thrombin affinity exhibited a strongly prometastatic phenotype, suggesting TM is a powerful determinant of hematogenous metastasis [101]. The antitumor effects of TM depend also on the activation of PC. In a mouse model of pancreatic cancer, recombinant TM interfered with tumor growth through inhibiting NF-κB and thrombin-induced PAR-1 activation [102]. There is also evidence that TM expression at the surface of tumor cells correlates with better prognosis in patients suffering from bladder, breast, colon, prostate, lung or oral epithelium cancer [103,104].

Anticoagulation is an effective process that involves a large panel of molecular players. Tumor cells may serve as prothrombotic triggers through aberrant regulation of receptor expression. Therefore, anticoagulation of cancer patients via direct oral anticoagulants or low-molecular-weight heparins appears to be a valid strategy to counteract hypercoagulation [105,106]. Heparins are close relatives of HS, which is an abundant glycan expressed by every mammalian cell including cancer cells. To which extent the HS of cancer cells has the potential to interfere with the coagulation system is discussed in the following section.

## 5. Heparan Sulfate

HS is a glycan covalently connected to a protein backbone forming so-called heparan sulfate proteoglycans (HSPG). HSPGs are ubiquitously present in mammals and exist as soluble or membrane-bound proteins. Membrane-bound proteins which are, thus, exposed at cell surfaces are glypicans and syndecans. While glypicans are connected to the plasma membrane via glycophosphatidylinositol anchors, syndecans are transmembrane proteins. The synthesis of the glycan chains of proteoglycan takes place within the Golgi apparatus and involves the action of various enzymes [107]. After the elongation of the HS chain by exostosins, several sulfotransferases introduce sulfate groups at different molecular positions. These modifications are not obligate but depend on the expression of the acting sulfotransferases and the availability of the required substrates [108,109]. Accordingly, the HS chain exhibits vast structural diversity, which is also reflected by the long list of biological properties and molecular interaction partners [110,111]. The ability of HS to bind human plasma proteins has previously been explored via pull-down assays coupled to a mass-spectrometry-based analysis [112]. In total, 99 plasma proteins with HS-binding properties have been identified. However, only a few studies investigated the (patho)physiological relevance of this interaction.

HS is a strongly negatively charged carbohydrate, and the binding of proteins depends at least in part on electrostatic interaction. Previously, Cardin and Weintraub described the amino acid consensus sequences that are involved in electrostatic interactions [113]. This so-called “Cardin Weintraub” motif contains the basic amino acids arginine or lysine in close proximity. Cardin and Weintraub postulated the HS binding site as one connected linear stretch of amino acids. However, insights that are more recent revealed that the spatial proximity of the required amino acids due to a specific three-dimensional configuration of the protein creates an HS binding cleft with high affinity [114]. Accordingly, binding affinities and specificities depend on the conformation of the HS binding site of the protein [115]. However, the structure of HS also defines this interaction. For instance, the binding of AT to HS is favored by a concise pentasaccharide sequence, which also inspired the synthesis of fondaparinux, a clinically relevant anticoagulant [116]. Recent studies showed that enhanced 3-O-sulfation of HS promoted the binding of AT to cancer cells [117].

AT, also known as AT III, is the main inhibitor of coagulation next to APC. As a plasmatic protein, AT has the ability to bind to the HS of CTCs, preventing the activation of thrombin and FXa. Once bound to HS, AT undergoes a conformational change. This enhances its reactivity towards thrombin, thereby promoting the formation of an AT/thrombin complex. Upon binding to HS, AT showed a strongly increased activity in neutralizing the enzymatic activity of thrombin [117,118,119]. Moreover, HS can interact with TFPI, which is a potent inhibitor of the blood coagulation factors FXa and FVIIa, counteracting coagulation [120,121].

The role of HS in coagulation is complex, since besides being anticoagulant, HS may also exhibit pro-coagulant properties. The interaction of HS with fibrinogen has been documented to reduce anticoagulant activity through forming an HS–fibrinogen–thrombin ternary complex, reducing the capacity of HS to bind to AT [122]. Moreover, and as previously mentioned, fibrinogen is recruited to the cancer cell surface by integrins. Interestingly, the activity of integrins is at least partially regulated by cell-surface-exposed HS, indicating complex and interwoven communication between the HS exposed by CTCs and the coagulation system [123].

In addition to its well-established anticoagulant function, AT has also been reported to exert anti-tumor properties. The binding of AT to cell-surface HS competes with the binding of growth factors blocking their proangiogenic effects [124]. AT can also attenuate inflammatory events through inhibiting NF-κB. Interestingly, the inhibitory potency of AT depends on the interaction of AT with cell-surface HS [125]. Similarly, it was reported that AT could also act as a modulator of tumor cell migration and invasion in different tumor cells. The inhibitory process required the activation of AT by HS [126]. Figure 4 illustrates the dual functions of HS in coagulation and its anti-tumor effects through AT binding.

The crosstalk between AT and HS is a prominent example reflecting the relevance of HS in regulating coagulation. However, similar effects can be expected for other plasma proteins. We have recently shown that HS exposed by CTCs together with αvβ3 integrins triggers the binding of vWF to the cell surface [108] (Figure 5). VWF is a complex multimeric plasma glycoprotein critical for normal hemostatic function. VWF has two main roles in hemostasis: first, to recruit and tether platelets at sites of vascular injury, facilitating platelet aggregation. Second, vWF acts as a protective carrier molecule for procoagulant FVIII. The hemostatic function of these vWF multimers is tightly regulated by the vWF-specific protease ADAMTS13 (a disintegrin and metalloproteinase with thrombospondin type 1 motif, 13). Interestingly, thrombin inhibits ADAMTS13, which further amplifies platelet recruitment and coagulation. Endothelial cells release vWF constitutively into the plasma at low levels. Increased secretion of vWF is followed upon endothelial stimulation with thrombin or tumor-cell derived vascular endothelial growth factor A [127,128]. This is in line with clinical studies which showed elevated vWF levels in the plasma of cancer patients. Together with a reduced plasma concentration of ADAMTS13, more plasmatic vWF was associated with a higher risk of developing venous thromboembolism [129].

We found that the abundance and length of the cell-surface-exposed HS chain coordinate the recruitment of plasmatic vWF to the cancer cell surface, which blocked, e.g., the binding of platelets. Murine tumor models and analysis of patients’ samples indicated that loss of this anti-coagulative HS at the cancer cell surface increased the ability of the CTC to metastasize, which was in turn connected to reduced patients’ survival [108].

## 6. Conclusions

The communication of CTCs with their environment defines their ability to survive within the blood stream and to form metastasis. In the present review, we shed some light on the interaction of CTCs with plasma proteins. This interaction is mediated by various cell-surface-exposed receptors and triggers various intracellular signaling pathways as well as extracellular reactions such as blood coagulation. Which plasma proteins can be recognized by CTCs and to which extent this interaction affects the fate of the CTC are largely unknown and comprehensive research is still missing. In this review article, we aimed to emphasize that improved knowledge of the CTC interactome is relevant to understanding the strategies of CTCs to survive and metastasize. However, the crosstalk between CTCs and plasma proteins is a double-edged sword. While the coagulation system is often considered to be exploited by the cancer cells, they are also a relevant part of the innate immune system and, as such, a powerful weapon to eliminate CTCs. This complex situation is also reflected in clinical data. Although the positive correlation between CAT and tumor progression suggests the coagulation system to be tumor supportive, therapy of cancer patients with anticoagulants limits successful thrombosis but has most often no direct tumor suppressive effects. Therefore, we assume that some interactions with plasma proteins support CTCs, while others have the potential to hinder further cancer progression. The exact biological impact of each molecular interaction depends most likely on the cancer entity, the stage of the disease and on the genetic repertoire of individual CTCs. The detection and evaluation of CTCs have become valuable tools in diagnostics to monitor therapy and patients’ outcomes. In recent years, the development of single-cell analysis techniques such as single cell RNA sequencing enable the in-depth characterization and advanced classification of CTCs. Further insights into the biological meaning of gene or protein expression patterns might be possible through investigating the interaction of single CTCs with blood components such as plasma proteins. Currently, there is a lack of studies that specifically describe methodologies to analyze the interactions between CTCs and plasma proteins in a clinical setting. CTC detection and isolation from patients’ blood is based on different technologies such as CTC enrichment through surface markers (i.e., epithelial cell adhesion molecule, EpCAM) or biophysical properties (i.e., size, density) [130,131,132,133]. The identification of CTCs in the blood of patients may also offer the opportunity to detect CTC-bound plasma proteins. Applicable methods may range from immunofluorescence microscopy and flow cytometry to single-cell mass spectrometry. These could not only improve our understanding of cancer metastasis but may allow better detection of cancer cells in liquid biopsy and differentiation between malignant and benign CTCs. Research has shown that TF-targeting therapeutics can effectively eradicate cancer stem cells isolated from breast, lung and ovarian cancer without the development of drug resistance. Given the high expression of TF in CTCs, this approach may also hold promise for the treatment of CTCs and metastatic disease. Overall, a better understanding of the CTC interactome and its impact on cancer progression is essential for the development of more effective diagnostic and therapeutic strategies for cancer patients [134].

## Figures and Tables

**Figure 1 cancers-15-03025-f001:**
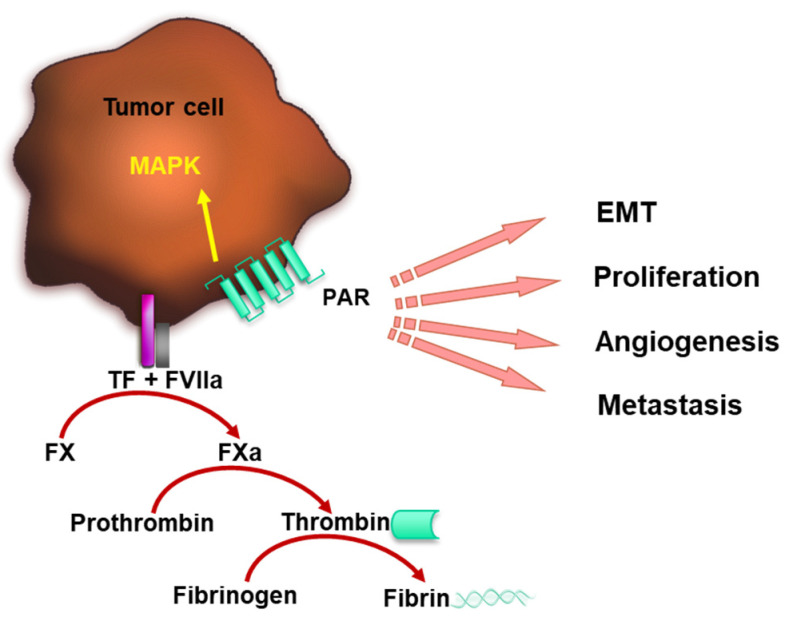
TF–FVIIa complex promotes coagulation and tumor progression. TF forms a complex with FVIIa catalyzing FX activation, which leads to thrombin formation. Thrombin cleaves fibrinogen to fibrin and contributes to blood clotting. FVIIa bound to the tumor cell surface by TF mediates PAR signaling, which induces MAPK pathway activation, tumor cell EMT, proliferation, angiogenesis and metastasis.

**Figure 2 cancers-15-03025-f002:**
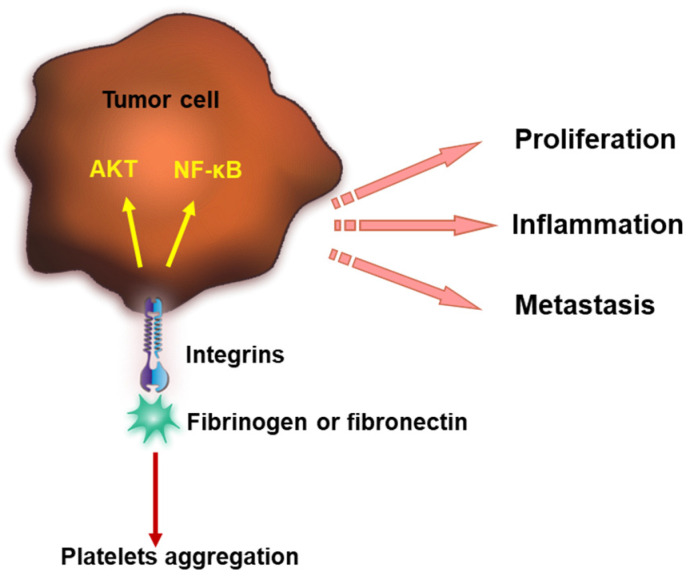
Integrins interact with fibrinogen or fibronectin, inducing platelet aggregation and coagulation and further contributing to tumor progression. Integrins bind fibrinogen or fibronectin and lead to AKT signaling and NF-κB activation. This promotes tumor proliferation, inflammation and metastasis.

**Figure 3 cancers-15-03025-f003:**
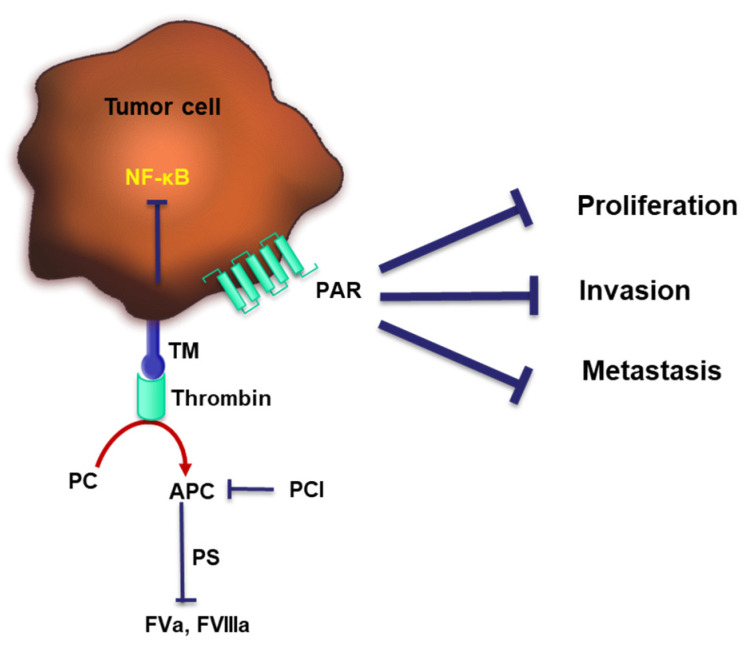
TM switches thrombin from a pro-coagulant to an anti-coagulant and prevents tumor progression. TM interacts with thrombin to subsequently activate PC. Activated PC (APC) in concert with PS switch on the PC anti-coagulant pathway. Thrombin can directly activate PAR-1, whereas the complex of thrombin and TM can active NF-κB, both pathways were shown to exert tumor-suppressing properties through dampening cell proliferation, invasion and metastasis.

**Figure 4 cancers-15-03025-f004:**
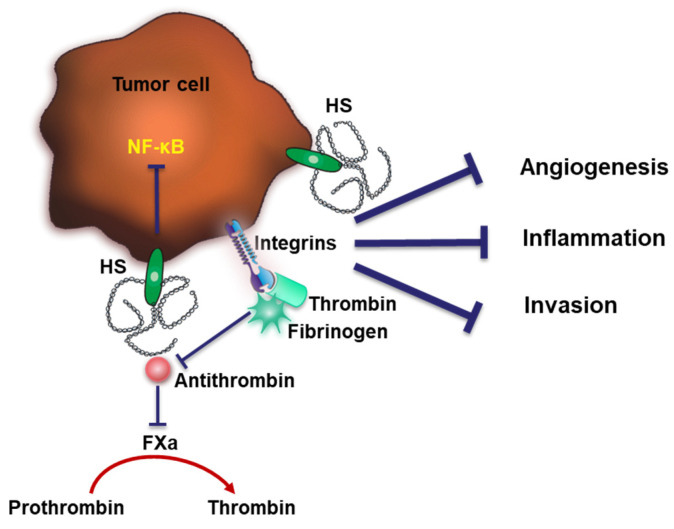
Dual functions of HS in coagulation. Cell-surface HS interacts with AT and neutralizes thrombin to inhibit coagulation. HS may also exhibit pro-coagulating capacities through forming a complex with fibrinogen and thrombin. The binding of AT to HS was shown to exert anti-tumor effects through inhibiting tumor angiogenesis, inflammation and invasion.

**Figure 5 cancers-15-03025-f005:**
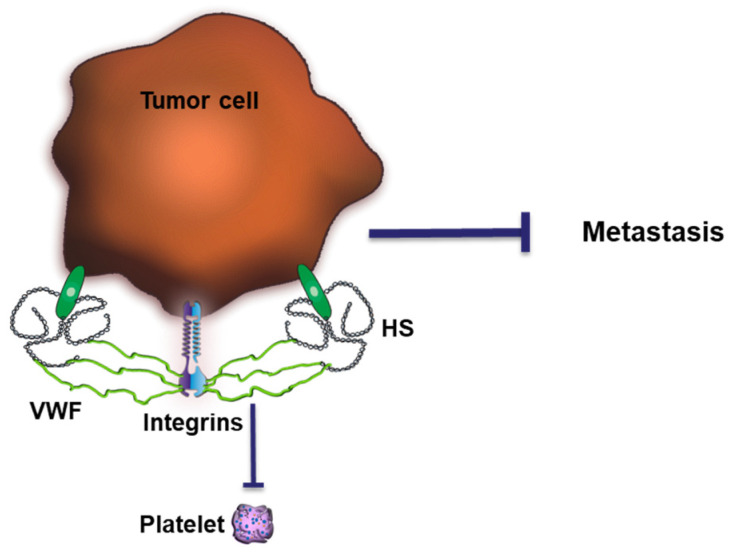
Impact of HS-mediated binding of vWF on tumor progression. VWF forms a complex with tumor cell surface HS and integrins, preventing platelet binding and hematogenous metastasis.

## Data Availability

Not applicable.

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
