# Peer review of "Crosstalk between Circulating Tumor Cells and Plasma Proteins—Impact on Coagulation and Anticoagulation"

_cancers, 2023, doi:10.3390/cancers15113025_

Round 1

Reviewer 1 Report

The review is an important and useful contribution to the field. It is well-written, with very good and clear figures summarizing major interactions. To make it even better and more comprehensive, I would request updates/additions on two topics:

1. Authors rightfully list the cancer-associated thrombosis (CAT) as a major problem in cancer patients, potentially CTC related.  Two older works on CAT are cited. However there have been more recent works on CAT-CTC link and they are worth citing and discussing briefly. For example: doi.org/10.1161/ATVBAHA.122.318715, or doi.org/10.3802/jgo.2013.24.3.265, but there are even more works.  

2. It will be worth to devote at least a few sentences on the link between CTC-plasma proteins interactions and epithelial-mesenchymal transition (EMT). EMT is considered one of major drivers of invasiveness and its role in CTCs is studied extensively. There are multiple links relevant to the topic of the review, for example a tissue factor expression on CTCs is related to EMT, and thrombin induces EMT. Please, add a relevant discussion.

Minor: in the Abstract “circulating cancer cells” are abbreviated as “CTCs”; obviously it should be “circulating tumor cells” to avoid confusion.

Author Response

We thank the reviewer for the detailed evaluation of our work and are delighted by the positive feedback.

Specific comments

Point 1: Authors rightfully list the cancer-associated thrombosis (CAT) as a major problem in cancer patients, potentially CTC related. Two older works on CAT are cited. However, there have been more recent works on CAT-CTC link and they are worth citing and discussing briefly. For example: doi.org/10.1161/ATVBAHA.122.318715, or doi.org/10.3802/jgo.2013.24.3.265, but there are even more works. 

Response 1: We agree with the reviewer. Therefore, in the revised manuscript we added further literatures [26, 27], [30-32] (page 2, lines 82-86, 96) and a brief discussion (page 2, lines 96).

Point 2: It will be worth to devote at least a few sentences on the link between CTC-plasma proteins interactions and epithelial-mesenchymal transition (EMT). EMT is considered one of major drivers of invasiveness and its role in CTCs is studied extensively. There are multiple links relevant to the topic of the review, for example a tissue factor expression on CTCs is related to EMT, and thrombin induces EMT. Please, add a relevant discussion.

Response 2: We thank the reviewer for raising this point. We added details on EMT in the text and figure 1 (page 3, lines 137-141). We added: “…through activating PARs and mitogen-activated protein kinase (MAPK) pathway promoting e.g. epithelial-mesenchymal transition (EMT) (Figure 1). EMT is a key process that enhances the ability of CTCs to disseminate as it drives vascular extravasation. Therefore, molecular markers of EMT and related signaling pathways may predict the ability of CTCs to metastasize. [41, 45-48] In this context, it has been shown that...”. Additionally, we adapted Figure 1.

Point 3: Minor: in the Abstract “circulating cancer cells” are abbreviated as “CTCs”; obviously it should be “circulating tumor cells” to avoid confusion.

Response 3: We followed the suggestions of the reviewer (page 1, line 17).

Reviewer 2 Report

I would first like to congratulate you for the extensive literature review on the subject. Targeting the cells of the blood circulation to either prevent the formation of CTC clusters or their interactions with single CTCs in the circulation may lead to a significant delay in metastatic potential and thus increased patient survival.One question that may not be answered is the appropriate methodology to be able to analyze these interaction studies with a clinical application. What could be such approaches?

Author Response

We highly appreciate the comprehensive evaluation and constructive feedback of the reviewer.

Specific comment: One question that may not be answered is the appropriate methodology to be able to analyze these interaction studies with a clinical application. What could be such approaches?

Response: As suggested, we discussed appropriate methodologies to analyze the CTC-plasma protein interaction (page 9, lines 374-381). In the revised manuscript we wrote: “Currently, there is a lack of studies that specifically describe methodologies to analyze the interactions between circulating tumor cells (CTCs) and plasma proteins in a clinical set-ting. CTC detection and isolation from patients’ blood based on different technologies such as CTC enrichment through surface markers (i.e., epithelial cell adhesion molecule, EpCAM) or biophysical properties (i.e., size, density) [129-132]. The identification of CTCs in the blood of patients may offer also the opportunity to detect CTC bound plasma proteins. Applicable methods may range from immunofluorescence microscopy, flow cytometry to single cell mass spectrometry”.

Reviewer 3 Report

Line 17:  Circulating cancer cells, should be "circulating tumor cells" as written on Line 8. 

Line 34: from "a".  Line 35: presence "of" 

Lines 49-57: The authors discuss several aspects of the metastatic process, and mention "accumulating studies", yet refer to only one reference,10.  "What happened to the other "several studies".  Similarly from lines 84-89.

Line 98: The "Interactions" between CTC ...... Also another suggestion, at the end of Line 111 can be added, "these interactions are described in detail below".

Line 126: Did the authors intend to use the word "tenase". Explain

Figures:  I believe that Tumor cell is represented well. Each process is described in the Figure legend.  However, I would suggest that the components of the various interactions should be labeled on the figures. That would be helpful fo the reader.

Line 243: Tumor cells may server as, should be "Tumor cells mat serve as a"

Line 272: ") patho)" rather than pato.  Line 329: clinical studies "which" showed.  Line 335:  "patients" samples.  Line 340: "hematogenous"   Line 351: "metastasize" 

Author Response

We appreciate the detailed feedback of the reviewer. We made corrections in the review accordingly.

Specific comments:

Point 1: Line 17:  Circulating cancer cells, should be "circulating tumor cells" as written on Line 8.

Response 1: We followed the suggestions of the reviewer (page 1, line 17).

Point 2: Line 34: from "a".  Line 35: presence "of"

Response 2: We followed the suggestions of the reviewer (page 1, lines 34, 35).

Point 3: Lines 49-57: The authors discuss several aspects of the metastatic process, and mention "accumulating studies", yet refer to only one reference,10.  "What happened to the other "several studies".  Similarly from lines 84-89.

Response 3: We followed the suggestions of the reviewer and added additional references (page 2, lines 51-53, 82-86).

Point 4: Line 98: The "Interactions" between CTC ...... Also another suggestion, at the end of Line 111 can be added, "these interactions are described in detail below".

Response 4: We followed the suggestions of the reviewer (page 3, lines 99, 112-113). We wrote “ In the following sections, these interactions are described in detail ”.

Point 5: Line 126: Did the authors intend to use the word "tenase". Explain

Response 5: We modified the related sentence. In the revised manuscript we wrote: “…TF-FVIIa complex…” instead of “…TF-FVIIa extrinsic tenase complex…” (page 3, lines 127-128).

Point 6: Figures:  I believe that Tumor cell is represented well. Each process is described in the Figure legend.  However, I would suggest that the components of the various interactions should be labeled on the figures. That would be helpful for the reader.

Response 6: We thank the reviewer for the suggestion and modified the figures 1-5, to enhance their clarity and improve visual presentation.

Point 7: Line 243: Tumor cells may server as, should be "Tumor cells may serve as a"

Response 7: We followed the suggestions of the reviewer (page 6, line 249).

Point 8: Line 272: "(patho)" rather than pato.  Line 329: clinical studies "which" showed.  Line 335:  "patients" samples.  Line 340: "hematogenous"   Line 351: "metastasize"

Response 8: We followed the suggestions of the reviewer (lines 279, 335, 341, 346, 357).